# Damping Scenarios of Kink Oscillations of Solar Coronal Loops

**Valery M. Nakariakov** [1,2,*,†] 🆔 and **Naga V. Yelagandula** [3,†] 🆔

1    Centro de Investigacion en Astronomía, Universidad Bernardo O'Higgins, Avenida Viel 1497, Santiago 8370993, Chile
2    Saint Petersburg Branch, Special Astrophysical Observatory of RAS, 196140 Saint Petersburg, Russia
3    Special Astrophysical Observatory of RAS, 369167 Nizhny Arkhyz, Russia
*    Correspondence: vnakariakov@googlemail.com
†    These authors contributed equally to this work.

**Abstract:** The transition from the large-amplitude rapidly-decaying regime of kink oscillations of plasma loops observed in the corona of the Sun to the low-amplitude decayless oscillations is modelled. In this study, the decayless regime is associated with the energy supply from coronal plasma flows, i.e., self-oscillations, or random movements of footpoints of the oscillating loop. The damping is attributed to the linear effect of resonant absorption. We demonstrate that the decay of an impulsively excited kink oscillation to the self-oscillatory stationary amplitude differs from the exponential decay. The damping time is found to depend on the oscillation amplitude to the power of a negative constant whose magnitude is less than unity. In this scenario, a better model for the damping seems to be super-exponential. In the separately considered case of the decayless oscillatory regime supported by a random driver, the oscillation amplitude experiences an exponential decay to the decayless level. Implications of this finding for magnetohydrodynamic seismology of the solar corona based on the effect of resonant absorption are discussed.

**Keywords:** magnetohydrodynamic waves; solar corona; magnetohydrodynamic seismology

## 1. Introduction

Magnetohydrodynamic (MHD) wave processes in the corona of the Sun have attracted growing attention for several decades [1]. The waves are considered as possible energy carriers in the context of the enigmatic problem of coronal heating, e.g., [2] and also as natural probes of plasma in the technique of MHD seismology, e.g., [3–6]. An intensively studied coronal MHD wave phenomenon is the kink mode oscillation of coronal plasma loops, see, e.g., [5] for a recent review. Kink oscillations are standing fast magnetoacoustic waves guided by a perpendicular non-uniformity of the fast speed, performing transverse displacements of the loop. Typical kink oscillation periods are several minutes. Specific values of the oscillation periods are determined by the length of the oscillating loop and the so-called kink speed linked with the Alfvén speeds inside and outside the loop [7]. This dependence is unequivocally confirmed by the linear scaling of kink oscillation periods with loop lengths, determined empirically with the use of high-resolution EUV imagers [8–10].

Kink oscillations appear in two regimes. In the large-amplitude rapidly decaying regime, displacement amplitudes reach several Mm, i.e., several minor radii of the oscillating loop [10]. Typically, the oscillations are excited by an initial displacement of the loop by a low coronal eruption [11]. In this regime, the oscillations decay very rapidly, in a few or several oscillation cycles. The oscillation damping is interpreted as the effect of resonant absorption, i.e., linear coupling of collective kink oscillations with torsional Alfvén oscillations of individual surfaces of a constant Alfvén speed, see, e.g., [12–14]. Resonant absorption could be considered as a linear energy cascade to progressively smaller radial scales. According to the resonant absorption theory, the damping profile is either exponential or Gaussian in different phases of the oscillation [15–17]. Specifically, the

damping has a Gaussian profile in early stages of the decaying oscillation if the density contrast in the oscillating loop is low. There are also nonlinear models of the kink oscillation damping, based on the Kelvin–Helmholtz instability of the loop's boundary or the resonant layer, e.g., [18–20]. This damping mechanism can be considered as a nonlinear energy cascade to smaller radial and azimuthal scales. The oscillation damping could be quantified by the quality factor defined as the ratio of the damping time estimated by an exponential fitting of observed decaying oscillation patterns and the oscillation period. Goddard and Nakariakov [21] demonstrated the power-law dependence of the kink oscillation quality factor upon the initial amplitude to the power about $-2/3$ (see also, [5]). Recently, Van Doorsselaere et al. [22] demonstrated theoretically that the quality factor caused by the nonlinear development of the Kelvin–Helmholtz instability (KHI) scales with the oscillation amplitude to the power of minus one, which is very close to the empirically determined dependence.

MHD seismology of solar coronal loops by decaying kink oscillations is mainly based on the use of two observables determined in the time domain: the oscillation period and the damping time (e.g., [23–25]). In more advanced models, an additional observable is the switch time from the Gaussian to exponential damping patterns [25–27]. The theoretically predicted exponential and Gaussian damping profiles used in the determination of the damping time have been derived under the assumption that the oscillation decays to zero. However, when the same loop oscillates successively in the decaying and decayless regimes, the damping profile should be determined accounting for the amplitude decrease to the saturated level.

In the other, decayless regime, the displacement amplitude is typically smaller than one Mm, i.e., is a fraction of the minor radii of the oscillating loop [8]. In some cases, larger amplitudes, up to 1.16 Mm, have been detected [28]. Similar oscillations have recently been detected in small loops in coronal bright points [29]. The decayless kink oscillations are not related to coronal eruptions and flares, and hence can be considered as a typical dynamic process of the quite-Sun period of time. In the decayless regime, the amplitude remains of the same order of magnitude for several tens of oscillation cycles, experiencing smooth irregular variations up and down [30–34]. Neither the amplitude nor the oscillation period show systematic evolution. Similarly, parameters of the oscillating loop, such as the minor radius, brightness in the observational bandpass, and steepness of the perpendicular profile do not show a systematic variation. In other words, in certain time intervals, the oscillation system seems to return to the initial state [33]. This apparent reversibility of the parameters of the decayless oscillation and of the oscillating loop raises a question of whether the oscillation damping is caused by KHI which is an irreversible process. In different periods of time, the same loop could oscillate in different regimes, while the oscillation period remains the same [28,35], or gradually changes with the evolution of the length of the oscillating loop [36], indicating that the physical processes in those two regimes differ by the amplitude only. The ubiquity of decayless kink oscillations suggests that in the cases when the oscillation amplitude decreases apparently to zero, it actually approaches the decayless level, which is missing from observations because of the insufficient resolution.

There are several mechanisms suggested to maintain the quasi-steady oscillation amplitude, i.e., allowing for the existence of the decayless regime. The oscillation energy losses by resonant absorption or another damping mechanism could be counteracted by random footpoint movements, e.g., [37,38] or the interaction of the loop with quasi-stationary coronal flows, e.g., [39–41], or their combination [42]. In both these mechanisms, the oscillation period is the period of the standing kink mode, which is very important for seismological applications [43]. There are also alternative suggestions which link decayless kink oscillations with interference fringes [44] and the developed state of KHI [45]. In addition, a decayless phase has been seen in ideal MHD without an external energy supply [17], while its nature requires further investigation.

The aim of this study is to determine the time damping profiles of impulsively excited kink oscillations, accounting for the decayless regime. In other words, we consider the

case when an impulsively decaying oscillation decreases not to zero, but to a stationary amplitude. The paper is organised as follows. In Section 2, we present the model. Section 3 describes results of the calculations, which are summarised and discussed in Section 4.

## 2. The Model

Following [39,42], we consider kink oscillations in terms of a zero-dimensional model based on a decaying simple harmonic oscillator driven by external regular and random flows,

$$\ddot{\xi} + \delta\dot{\xi} + \Omega_k^2\xi = F(v_0 - \dot{\xi}) + N(t), \tag{1}$$

where $\xi$ is the oscillation amplitude (i.e., the transverse displacement of the loop), $\delta$ is a constant which describes damping by, e.g., resonant absorption, $\Omega_k$ is the natural frequency of a standing kink wave, determined by the loop's length $L$ and the kink speed $C_k$ as $\Omega_k = \pi C_k / L$; and the force $F$ describes the interaction of the loop with an external coronal flow with the speed $v_0$. In addition, the oscillation could be driven by random motions at footpoints, which are accounted by the term $N$ which is a function of time $t$.

Accounting for first two terms in the Taylor expansion of the function $F$ by the speed $\dot{\xi}$, we obtain a randomly driven Rayleigh oscillator equation,

$$\ddot{\xi} + \left[(\delta - \delta_v) + \alpha\left(\dot{\xi}\right)^2\right]\dot{\xi} + \Omega_k^2\xi = N(t), \tag{2}$$

where the constant parameters $\delta_v$ and $\alpha$ are the coefficients of the two lowest order terms in the expansion (e.g., [46]). The combination of the positive and negative linear damping coefficients determines the behaviour of the low-amplitude perturbation. In the absence of noise, $N(t) = 0$, Equation (2) reduces to the standard Rayleigh oscillator equation. For positive $\delta - \delta_v$, the oscillation is growing, while for the negative one, it is decaying. For negative $\delta - \delta_v$ and positive $\alpha$, the Rayleigh equation has the stable limit cycle solution

$$\xi_\infty = \sqrt{4|\delta - \delta_v|/(3\alpha\Omega_k^2)}, \tag{3}$$

e.g., [47]. This solution corresponds to the decayless (i.e., stationary) oscillation with the period $2\pi/\Omega_k$ and the constant amplitude $\xi_\infty$. As the system is essentially dissipative and energetically open, this oscillatory motion is self-sustained, or a self-oscillation. We need to stress that if the coefficient $\alpha$ tends to zero, the stationary oscillation amplitude goes to infinity, which corresponds to the solution of a simple harmonic oscillator with the negative friction. The oscillation frequency is practically insensitive to multiplicative and additive noises in the system [42], i.e., the Rayleigh self-oscillation is "relatively immune against disturbances" [48]. The self-oscillatory solution was linked with the decayless regime of kink oscillations [39], which has been confirmed by full MHD numerical simulations [40].

If the initial amplitude $\xi_0$ is lower or greater than the stationary amplitude $\xi_\infty$, the oscillation amplitude experiences either a growth or decrease to $\xi_\infty$. The inspection of Equation (2) demonstrates that there is no reason the damping scenario to be exponential because of the presence of the nonlinear term. Moreover, even if the decay could be well fitted with an exponential function, the characteristic damping time $t_{exp}$ does not need to be equal to $t_{exp} = 2/\delta$, as it is in Equation (1) with $N(t) = 0$.

## 3. Results

Equation (2) supplemented by the initial conditions $\xi(0) = \xi_0$, $\dot{\xi}(0) = 0$ which imitate the excitation by a displacement of the loop by a low coronal eruption, is solved with the use of the *dsolve* function of the symbolic and numeric computing environment *Maple 2020.2*. The chosen solver scheme is the Fehlberg fourth-fifth order Runge–Kutta method with degree four interpolant, which is the default option in *dsolve*. We have taken the minimum time step $dt = 10^{-3}(2\pi/\Omega_k)$, i.e., one thousandth of the oscillation period, which for our purpose gives enough accuracy as can be seen by comparing the numerical results at large $t$ with the saturation value of the displacement $\xi_\infty$ given by Equation (3). In contrast with

the previous study [42], we consider the case $\xi_0 > \xi_\infty$, paying the main attention to the decrease in the oscillation amplitude to the stationary level. The decay pattern has been quantified by fitting the absolute values of the extremes of $\xi(t)$ with a guessed model function, using the *fit* function of the *Statistics* package of *Maple 2020.2*, which implements the least-squares error technique.

As the model decay functions, we used the exponential and super-exponential functions described by the following expressions, respectively:

$$\frac{\xi}{\xi_\infty} = 1 + \left( \frac{\xi_0}{\xi_\infty} - 1 \right) \exp \left\{ -\frac{t}{t_c} \right\}, \tag{4}$$

and

$$\frac{\xi}{\xi_\infty} = 1 + \left( \frac{\xi_0}{\xi_\infty} - 1 \right) \exp \left\{ -\left( \frac{t}{t_c} \right)^d \right\}, \tag{5}$$

where the constant $d$ is the super-exponential index. Both functions (4) and (5) have values $\xi_0$ at $t = 0$, and go to $\xi_\infty$ at $t \to \infty$. The choice of the super-exponential decay model, in addition to the standard exponential model, is motivated by the study [49]. In that work, the authors investigated damping of kink oscillations in the form given by Equation (5), and found $d = 0.4$, $d = 1.8$ and $d = 2.8$ in three specific decaying kink oscillation events occurring after flaring energy releases. It should be taken into account that the analysed events were observed with the EUV imager TRACE which had poorer time resolution and sensitivity than SDO/AIA used for the catalogues of kink oscillation events [10,11] and follow-up studies based on those catalogues.

In the following, we consider separately solutions to the initial value problem of the Rayleigh oscillator equation, Equation (2) with $N = 0$ and a randomly driven oscillator, Equation (1) with $F = 0$.

### 3.1. Case of Self-Oscillations

Figure 1 demonstrates typical decaying patterns of impulsively excited self-oscillations, described by Equation (2) with $N = 0$, with different initial amplitudes and values of the combined damping coefficient $\delta - \delta_v$. In all cases shown in the figure, the initial amplitude is higher than the stationary amplitude $\xi_\infty$. The oscillations experience decay, gradually approaching $\xi_\infty$. Visually, the oscillation envelope decreases exponentially, which allows us to determine the exponential decay time $t_{ex}$. Tables 1 and 2 show the best fitting parameters for the damping model given by Equation (4) for different initial amplitudes and combinations of the positive and negative linear damping coefficients $\delta - \delta_v$. It is evident that in the exponential damping fitting, the characteristic time $t_{ex}$ depends on the initial amplitude, decreasing with the increase in the amplitude. This dependence is caused by the nonlinear nature of the governing Equation (2). For $\delta - \delta_v = -0.1$, the quality factor of the oscillation, defined as $Q = t_{ex}/(2\pi/\Omega_k)$, scales with the initial amplitude as $Q \propto \xi_0^{-1.1}$. For $\delta - \delta_v = -0.5$, the scaling is $Q \propto \xi_0^{-0.8}$.

According to Figure 1, the oscillation damping patterns are not fitted by the exponential damping model sufficiently well. The fitting errors are rather large, reaching about 50%, see Tables 1 and 2. A better fit is achieved with the super-exponential damping function (5). The best-fitting super-exponential index $d$ is typically less than unity. As well as in the exponential damping scenario, the characteristic super-exponential damping time, $t_{sup}$, decreases with the increase in the initial amplitude. For $\delta - \delta_v = -0.1$ and $-0.5$, the quality factor scales as $Q \propto \xi_0^{-0.6}$ and $Q \propto \xi_0^{-0.7}$. On the other hand, for the combinations of the parameters given in the last two rows of Table 2, the super-exponential fit has a bigger error than the exponential fit. It indicates that neither of those fitting functions are sufficiently good, and there may be a need for another one. However, currently, we do not have any reason motivated by the theory that gives a clue about a better analytical fitting function for the damping scenario; therefore, the super-exponential fitting seems to be a suitable choice.

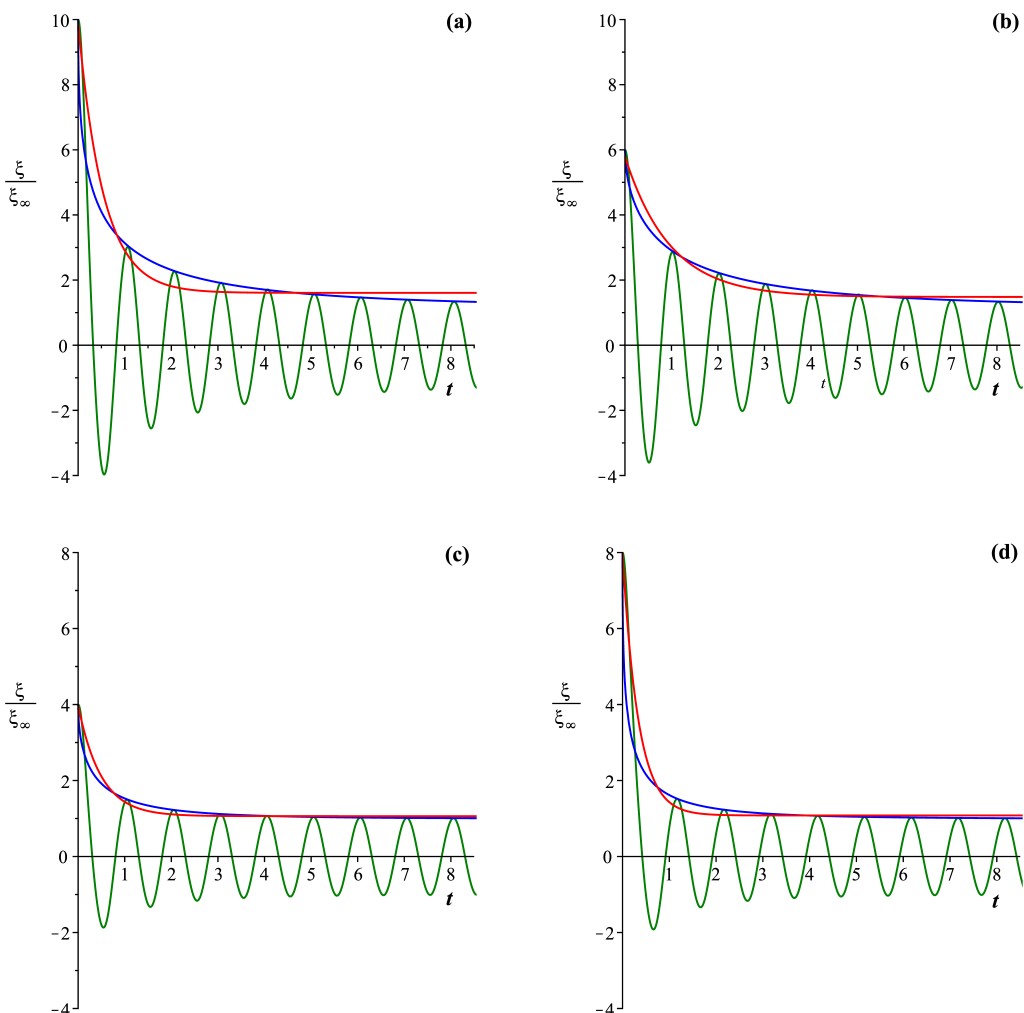

**Figure 1.** Evolution of an impulsively excited self-oscillation with the initial amplitude exceeding the stationary amplitude (green). The red and blue curves show the exponential and super-exponential fittings for the oscillation envelope, respectively. The oscillation amplitude is normalised to the stationary amplitude. The time is measured in oscillation periods, i.e., $\frac{2\pi}{\Omega_K}$. The parameter $\alpha$ is 1.2 and $\Omega_K = 2\pi$. Panel (**a**): $\delta - \delta_v = -0.1$, and $\xi_0/\xi_\infty = 10$. Panel (**b**): $\delta - \delta_v = -0.1$, and $\xi_0/\xi_\infty = 6$. Panel (**c**): $\delta - \delta_v = -0.5$, and $\xi_0/\xi_\infty = 4$. Panel (**d**): $\delta - \delta_v = -0.5$, and $\xi_0/\xi_\infty = 8$.

**Table 1.** Best-fitting damping times $t_{\text{ex}}$ and $t_{\text{sup}}$ in the exponential and super-exponential damping models, respectively, and the the super-exponential index $d$ for different normalised initial amplitudes $\xi_0/\xi_\infty$, and the respective fitting errors for the solutions to the Rayleigh oscillator equation with $\alpha = 1.2$, $\delta - \delta_v = -0.1$, $\Omega_K = 2\pi$.

| | **Exponential Fitting** | | | **Super-Exponential Fitting** | |
|---|---|---|---|---|---|
| $\xi_0/\xi_\infty$ | $t_{\text{ex}}$ | Fitting Error in Percentage | $t_{\text{sup}}$ | $d$ | Fitting Error in Percentage |
| 2 | 3.47 | 1.39 | 3.94 | 0.873 | 0.42 |
| 4 | 1.58 | 11.47 | 1.69 | 0.69 | 2.50 |
| 6 | 0.97 | 25.03 | 0.96 | 0.57 | 5.10 |
| 8 | 0.68 | 36.17 | 0.60 | 0.49 | 8.48 |
| 10 | 0.54 | 43.16 | 0.40 | 0.44 | 12.27 |

**Table 2.** The same as in Table 1, but for $\delta - \delta_v = -0.5$.

| | Exponential Fitting | | Super-Exponential Fitting | | |
|---|---|---|---|---|---|
| $\zeta_0/\zeta_\infty$ | $t_{\text{ex}}$ | Fitting Error in Percentage | $t_{\text{sup}}$ | $d$ | Fitting Error in Percentage |
| 2 | 1.01 | 4.75 | 0.96 | 0.76 | 1.18 |
| 4 | 0.49 | 16.76 | 0.37 | 0.55 | 7.85 |
| 6 | 0.37 | 21.01 | 0.19 | 0.47 | 15.85 |
| 8 | 0.34 | 22.25 | 0.13 | 0.43 | 22.60 |
| 10 | 0.32 | 22.91 | 0.10 | 0.41 | 28.11 |

### 3.2. Case of Randomly Driven Oscillations

Figure 2 demonstrates typical decaying patterns of impulsively excited harmonic oscillations described by the damped harmonic oscillation equation (Equation (1)) with $F = 0$. The driving term $N(t)$ is stationary red noise with a zero mean and the standard deviations of unity (cf., [42]), which represents aperiodic dynamic processes typical for the solar atmosphere (e.g., [50–52]). The red noise realisations were generated with the function *BrownianMotion* of the *Finance* package in *Maple 2020.2*. In our study, we solve numerically initial value problems for Equation (1) with the deterministic right-hand side, i.e., taking $N(t)$ as a specific realisation of the noise.

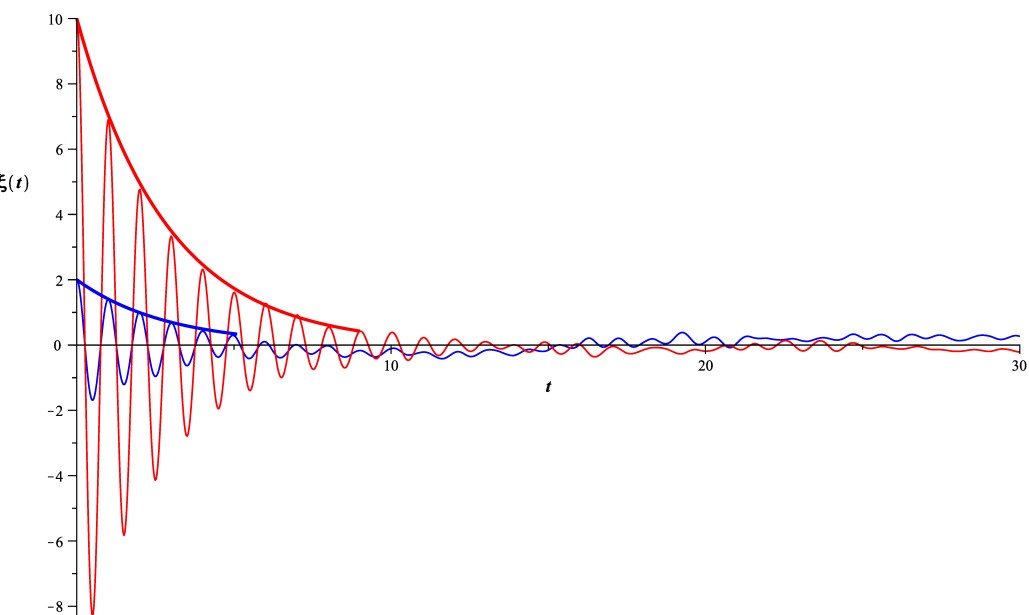

**Figure 2.** Evolution of impulsively excited oscillations described by a randomly driven harmonic oscillator with a linear damping term. The damping coefficient is $\delta = 0.7$. The red and blue oscillatory curves correspond to different initial amplitudes. The thick curves show the exponential damping of the oscillation envelopes with the characteristic time $2/\delta$.

An impulsively excited oscillation experiences decay, but does not cease to zero but keeps oscillating with the natural period $2\pi/\Omega_k$ (cf. [42]), forming a decayless oscillating pattern with a small amplitude. Such a behaviour resembles the observed evolution of a large-amplitude kink oscillation into a decayless oscillation [36]. The amplitude experiences some fluctuations which seem to be consistent with the observed behaviour too [33]. However, in this case, the damping time is independent of the oscillation amplitude, as the governing equation is linear.

## 4. Discussion and Conclusions

In this paper, we study various damping scenarios of an impulsively excited kink oscillation of a solar coronal loop to oscillations with a (quasi-)stationary amplitude, i.e., the transition from the large-amplitude rapidly decaying oscillatory regime to the low-amplitude decayless one. The main novel element of the study is the assumption that an impulsively excited kink oscillation decays to a decayless oscillation, and not to a zero amplitude as it has been assumed in a number of previous studies (e.g., [9,10,14,21,53]). The ubiquity of decayless oscillations suggests that the apparent decay to a zero amplitude could be caused by the insufficient spatial resolution of the observations. We considered separately two mechanisms for the decayless regime, a self-oscillation resulted from the interaction of the oscillating loop with a quasi-steady flow [39,40], and an oscillation which is sustained by random footpoint movements representing, e.g., the granulation [37,54]. The study was based on an initial value problem for a randomly driven Rayleigh oscillator equation, proposed and elaborated in [39,42]. So far, this approach has been applied to the study of the oscillation generation and sustaining, while the transition from the decaying regime to the decayless one is considered in our study for the first time.

In the self-oscillatory case, our results show that an impulsively excited oscillation with the initial amplitude exceeding the stationary amplitude $\zeta_\infty$ experiences an amplitude decay from the initial value to the stationary level. Thus, we can distinguish between two phases of the phenomenon, the decay phase, and the phase of the stationary self-oscillation, i.e., the decayless phase. In some cases, which were not addressed in our study, the initial part of the decay phase could be distinguished as the Gaussian decay phase (see, e.g., [17]).

In the decay phase, the damping pattern could be approximated with an exponential function, allowing for the estimation of the exponential damping time and oscillation quality factor. If the stationary amplitude $\zeta_\infty$ is smaller than the observational detection threshold, the amplitude decrease appears to be apparently consistent with the regular decay to the zero amplitude, described by resonant absorption. However, if the damping of the oscillation is the decay phase of a self-oscillation, the error associated with the exponential fitting is rather large, reaching 50%. This needs to be taken into account in the seismological techniques which use the exponential damping time as an observable, based on the assumption that the damping is caused by resonant absorption (e.g., [23–25,55]). We also experimented with the super-exponential fitting of the damping patterns, and found out that it has, generally, smaller errors than the exponential fitting. However, as the number of free parameters in the super-exponential function is larger than in the exponential one, comparison between the models requires Bayesian statistical analysis [25] which is out of scope of this paper. For the majority of considered combinations of the parameters, the super-exponential index is found to be less than one, in contrast with the value about two found observationally [49] in a limited set of oscillatory patterns. The super-exponential damping needs to be tested on observational data, e.g., using the catalogue presented in [10].

In addition, in the self-oscillatory case, the quality factor of the decay phase of the oscillation is found to decrease with the increase in the initial amplitude. The quality factor scales with the initial amplitude to the power about minus 1 or less. This result is consistent with the observationally established fact that the kink oscillation quality factor, determined empirically by the exponential model fitting, scales with the initial amplitude as $Q \propto \xi_0^{-2/3}$ [5,21]. On the other hand, if the damping is caused by resonant absorption which is a linear effect, the damping time should be independent of the initial amplitude value. In the absence of mechanisms sustaining the oscillation, in terms of the model described by Equation (1), the exponential damping time is determined by the effective damping coefficient $\delta$, i.e., is $2/\delta$. Obviously, the self-oscillatory and damped oscillation scenarios differ from each other by the amplitude at a long time after the initial excitation. However, if the self-oscillatory stationary amplitude is lower than the threshold of observational detection, one can readily confuse these two scenarios. Another explanation of the power-law dependence of the decaying kink oscillation quality factor on the initial amplitude is the nonlinear damping by KHI [22], which does not include the decayless regime in the model.

A similar two-phase behaviour occurs if the oscillator is driven by an external force randomly varying in time. In the decayless phase, the oscillation amplitude experiences gradual increases and decreases, which are consistent with the observed behaviour [33]. In contrast with the essentially nonlinear case of the self-oscillation, in the case with the random driver, the decayless regime is reproduced by a simple harmonic oscillator with a linear damping term. While in reality both mechanisms could operate simultaneously [42], it is important to know what are the minimal requirements for the occurrence of the decayless regime. In the case of a randomly driven simple harmonic oscillation with a linear damping, the decay pattern is well approximated by an exponential function. The exponential damping time $2/\delta$ and the quality factor are independent of the initial amplitude, justifying, in this case, the applicability of the seismological techniques based upon the damping time measurement.

The presented study demonstrates the power of the low-dimensional approach, giving sensible results consistent with observational findings. The developed theory assumes that a coronal loop behaviour could be described by the zero-dimensional model (2). As this model was constructed heuristically, its obvious shortcomings are the lack of the rigorous derivation of the governing equation, and the relationship of the model parameters $\delta_v$ and $\alpha$ with specific parameters of the coronal plasma. This relationship could be established by comparing outcomes of the model with results of numerical simulations of kink oscillations sustained by coronal flows and or random motions of footpoints, e.g., similar to described in [40]. However, one should be careful with the effect of artificial dissipation which is intrinsic to 3D numerical simulations of MHD processes. For example, simulated kink oscillations experience damping accompanied by the increase in the internal energy of the plasma even if the governing equations are the ideal MHD [56]. In terms of the zero-dimensional model based on a simple harmonic oscillator, this artificial dissipation modifies the damping coefficient $\delta - \delta_v$, resulting in additional damping (e.g., see the discussion in [57]). Another interesting next step in this study would be to include the nonlinear damping mechanism [22] into the low-dimensional model (1), and consider effects of the competition or co-existence of KHI, random driving, and self-oscillation with each other. In addition, despite 76% of the observed cases with decaying kink oscillations were induced by plasma eruptions [11], i.e., the oscillation began with a mechanical displacement of the loop from the equilibrium, the remaining events could be associated with another excitation mechanism. It would be of interest to investigate whether the parameter $d$ in the super-exponential damping scenario described by Equation (5) appears to be different in observed kink oscillations excited by different mechanisms.

Having dedicated our work primarily to the theoretical side of the decayless kink oscillations, we would like to conclude it by giving a very brief glimpse of the way observational data can be obtained specially to determine the damping profile of an impulsively excited kink mode due to a coronal eruption that later settles into a decayless phase. One of the problems with the observational data is the low time resolution of available EUV imagers. The radio emission that comes from the solar atmosphere, however, is mostly pristine and unaffected by the Earth's atmosphere and therefore can be detected by the radio telescopes on a continuous basis with a high temporal resolution. The big reflecting radio telescopes such as RATAN-600 and the radio interferometric telescopes, for example, e-OVSA, MUSER or SRH are well equipped from the perspective of spatial and spectral resolutions (e.g., [58]). In particular, kink oscillations have already been identified in the modulation of a flaring microwave emission [59]. Especially the 120 m to 200 m variable profile aperture of the RATAN-600 whose feed cabin has been retrofitted with a new tracking and sensitive spectral detection equipment can be used in the high-frequency radio spectrum to observe the decay phase of the kink mode oscillations, when the event takes place at the edge of the limb by using the well-known method of scintillation. The same method can be further used to get the actual stochastic variation of the coronal plasma flow near the oscillating coronal loop by observing a distant star or a supernovae remnant through the solar atmosphere and this information can directly be used in our models

instead of using a software-based random noise generator. However, we shall reserve this for our next work about the decayless kink oscillations.

**Author Contributions:** Conceptualization, V.M.N.; methodology, V.M.N.; software, N.V.Y.; validation, V.M.N. and N.V.Y.; formal analysis, V.M.N. and N.V.Y.; writing—original draft preparation, V.M.N.; writing—review and editing, V.M.N. and N.V.Y.; visualization and tables, N.V.Y. All authors have read and agreed to the published version of the manuscript.

**Funding:** This research was motivated by the need to lay down the theoretical framework for interpretation and analysis of the observational data that would be obtained from the modernized facility at RATAN-600 of SAO RAS and therefore is funded completely by the Ministry of Science and Higher Education of the Russian Federation as per the grant No. 075-15-2022-262 (13.MNPMU.21.0003).

**Data Availability Statement:** Not applicable.

**Conflicts of Interest:** The authors declare no conflict of interest.

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
