# Peer review of "Damping Scenarios of Kink Oscillations of Solar Coronal Loops"

_universe, doi:10.3390/universe9020095_

Round 1
Reviewer 1 Report
Please, find my comments in the attached PDF file.

Author Response
REVIEWER 1: 1.a. line 4: “The damping is attributed to the linear effect of resonant absorption” (Is this attributed by the scientific community or by the authors? Both of them?)
REVIEWER 1: 1.b. lines 7 and 8: “the decay of an impulsively excited kink oscillation to . . . . differs from the exponential decay typical for resonant absorption” (under what circumstances? In general? Do the authors attribute the damping to the resonant absorption or not?)
REVIEWER 1: 1.c. "lines 37 and 38: “According to the resonant absorption theory, the damping profile is either exponential or Gaussian in different phases of the oscillation”. (The exponential profile is, therefore, not typical of resonant absorption (?) )
REPLY: The abstract was rewritten according to these suggestions. Also, we complied with the 200-word limit.
REVIEWER 1: 2. "In line 66, the authors mentioned “the apparent reversibility of the decaying kink oscillation amplitude evolution”. Can the author provide more details about this reversibility?"
REPLY: The details are provided: "Neither the amplitude nor the oscillation period show systematic evolution. Similarly, parameters of the oscillating loop, such as the minor radius, brightness in the observational bandpass, or steepness of the perpendicular profile show a systematic variation. In other words, in certain time intervals the oscillation system seems to return to the initial state \citep{2022MNRAS.513.1834Z}. This apparent reversibility of the parameters of the decayless oscillation and of the oscillating loop raises} a question of whether the oscillation damping is caused by KHI which is an irreversible process."
REVIEWER 1: 3. "For the sake of clarity, can the authors specify how δv and α coefficients in equation (2) are related to the coefficients in equation (1)?"
REPLY: It is added after Eq. (2).
REVIEWER 1: 4. The authors in lines 113 and 114 mentioned “dsolve function” and “Maple 2020.2 environment”. Can they provide more information regarding the function mentioned above and the environment as soon as they are introduced?
REPLY: The information about the solver implemented in the function dsolve was added: "The chosen solver scheme is the Fehlberg fourth-fifth order Runge--Kutta method with degree four interpolant, which is the default option in \textit{dsolve}".
Also, it was added that Maple is a computing environment.
REVIEWER 1: 5. Please, cancel the comma at the beginning of line 123.
REPLY: Done, thank you.
REVIEWER 1: 6. In line 124. Can the authors specify the results of the study quoted in [48] that support their choice?
REPLY: This paper is now discussed in more detail in that paragraph: "In that work, the authors investigated damping of kink oscillations in the form given by Eq.~\ref{Fit_sup}, and found $d=0.4$, $d\approx 1.8$ and $d=2.8$ in three specific damped, kink oscillation events occurring after flaring energy releases. It should be taken into account that the analysed events were observed with the TRACE EUV imager which had poorer time resolution and sensitivity than SDO/AIA used for the catalogues of kink oscillation events \citep{2015A&A...577A...4Z, 2019ApJS..241...31N} and follow-up studies based on those catalogues."
REVIEWER 1: 7. Please, check the caption of table 2 (is this for δ = −0.5 or δ − δv = −0.5 ?)
REPLY: The typo was corrected, thank you!
REVIEWER 1: 8. Table 2. Can the authors motivate why the error for the super-exponential fitting is bigger than the exponential fitting in the last cases indicated in this table (i.e., the last two lines corresponding to ξ0/ξ∞ = 8 and 10)?
REPLY: We added a discussion of this issue: "On the other hand, for the combinations of the parameters given in the last two rows of Table~\ref{Tab:2}, the super-exponential fit has a bigger error than the exponential fit. It indicates that neither of those fitting functions are sufficiently good, and there may be a need for another one. However, as there is currently no a theoretically motivated alternative damping scenario, the super-exponential fitting seems to be a suitable choice."
Reviewer 2 Report
This manuscript presents a low-dimensional model for describing the behavior of kink oscillations in solar coronal loops, paying attention to how the system evolution transitions from a large-amplitude, rapidly decaying stage to a small-amplitude, decayless stage. The authors invoked two independent scenarios to attain the decayless stage, one being some organized ambient flow (the self-oscillation scenario) and the other being a random driver. The damping envelope was quantified, and the associated characteristics were examined in connection with the magnitude of the initial displacement.
Overall this manuscript is well motivated, the logical flow is clear, and the figures are well prepared. In particular, I believe that the strength of this study lies in the simplicity of the adopted model (Equation 1). I nonetheless have a number of minor comments for the authors to address before this manuscript is recommended for publication.
1. lines 10-11. The sentence “... a better model ... seems to be an exponential to the power of minus times to a power which ...” reads a bit awkward. Please consider rephrasing.
2. lines 35-37. “Resonant absorption could be considered as a linear energy cascade ...”. Please consider referencing 2015ApJ...803...43S (see references therein as well), which is particularly relevant here.
3. line 67, “rises a question of...” -> “raises”
4. lines 89-90, “...(e.g., the transverse displacement .. or the perpendicular speed)”. I would suggest to remove “or the perpendicular speed”, otherwise $\dot{\xi}$ is dimensionally incompatible with $v_0$.
5. Equation 2. I would suggest to briefly discuss the physical meanings of $\delta_v$ and $\alpha$ in the context of the function $F$. In particular, is it possible to seismologically deduce some information about $F$, supposing that $v_0$ can be known with some accuracy?
6. Figure 1. I would suggest the authors to mention that a transition from a rapidly decaying stage to an apparently decayless stage was also found in 2015ApJ...803...43S (see Figures 9 and 11 therein), where only linear physics is considered.
Author Response
REVIEWER 2: 1. lines 10-11. The sentence “... a better model ... seems to be an exponential to the power of minus times to a power which ...” reads a bit awkward. Please consider rephrasing.
REPLY: The abstract was rewritten according to this suggestion, and complying with the 200-word limit.
REVIEWER 2: 2. lines 35-37. “Resonant absorption could be considered as a linear energy cascade ...”. Please consider referencing 2015ApJ...803...43S (see references therein as well), which is particularly relevant here.
REPLY: The suggested paper is now referenced.
REVIEWER 2: 3. line 67, “rises a question of...” -> “raises”
REPLY: The typo is corrected, thank you!
REVIEWER 2: 4. lines 89-90, “...(e.g., the transverse displacement .. or the perpendicular speed)”. I would suggest to remove “or the perpendicular speed”, otherwise $\dot{\xi}$ is dimensionally incompatible with $v_0$.
REPLY: The "or the perpendicular speed" is deleted.
REVIEWER 2: 5. Equation 2. I would suggest to briefly discuss the physical meanings of $\delta_v$ and $\alpha$ in the context of the function $F$. In particular, is it possible to seismologically deduce some information about $F$, supposing that $v_0$ can be known with some accuracy?
REPLY: It is a very important question which could be reformulated as "what are the clear links of the model parameters $\delta_v$ and $\alpha$ with specific parameters of the coronal plasma?". We added some discussion of this issue to the Discussion section.
REVIEWER 2: 6. Figure 1. I would suggest the authors to mention that a transition from a rapidly decaying stage to an apparently decayless stage was also found in 2015ApJ...803...43S (see Figures 9 and 11 therein), where only linear physics is considered.
REPLY: We added the statement: "In addition, a decayless phase has been seen in ideal MHD without an external energy supply \citep{2015ApJ...803...43S}, while its nature requires further investigation.".
Reviewer 3 Report
Kink oscillations in the solar atmosphere are prevalent. In previous literatures, large-amplitude, fast-decaying and low-amplitude decayless kink oscillations are considered and investigated separately. For the first time, this manuscript self-consistently presents the transition from the large-amplitude rapidly-decaying regime of kink oscillations of plasma loops to the low-amplitude decayless oscillations. The damping scenarios in the two regimes are analyzed in detail. The results are of great significance and merit publication. I have only a few minor suggestions.
1. Are there observational evidences of obvious two regimes of loop oscillations? If yes, the authors may present just one case and apply the theory to observations. If no, please explain the reason. Is it related to the spatial resolution and time cadence of the instrument (e.g., SDO/AIA)? What are the technical requirements of next-generation space telescopes to make a breakthrough, especially in loop dynamics?
2. line 67: "rises a question" should be "raises a question"
The end.
Author Response
Thank you for the positive report.
- Yes, there are observational detections of the presence of both decaying and decayless kink oscillations regimes in the same loop. The relevant papers are cited in Introduction, "In different periods of time, the same loop could oscillate in different regimes, while the oscillation period remains the same \citep{2020A&A...638A..32Z, 2021A&A...652L...3M}, or gradually evolves with the evolution of the length of the oscillating loop \citep{2013A&A...552A..57N}, ". The re-analysis of those events with the use of the theoretical results of our manuscript, as well as the search for other similar events, should be carried out in a follow-up work.
- The typo is fixed, thank you!
Reviewer 4 Report
The authors addressed the origin of decayless oscillations in the solar corona, which may participate in the long-known problem of coronal heating.
This paper consists of a continuation of the research done by the group led by Prof. Valery Nakariakov. The authors developed further the former models and showed that an impulsively decaying oscillation decreases not to zero, but to a stationary amplitude. This is an important finding as such excited decaying oscillation may thermalize their energy and contribute to coronal heating.
The main novel element of the study is the finding that impulsively excited kink oscillations decay to decayless oscillations, and not to a zero amplitude as it has been found in a number of previous studies.
The methodology used by the authors in this paper is fully justified and it does not require to be improved. Their analytical work is supported by symbolic calculations performed with the use of the Maple 2020.2 environment and the obtained results are illustrated by Figs. 1 & 2 as well as by Tables 1 & 2.
The authors specified the aim of their studies at the end of Introduction. The obtained results are discussed and conclusions are correlated to the aim. Besides, the authors provide some information on improvement of their analytical model which further can consist of a nice background for a development of magnetohydrodynamic models.
55 references listed out at the end of the draft are fully justified and they are most relevant for the performed studies.
The draft is nicely written with most appropriate figs and tables. I have found only one misprint on p. 4, one line below Eq. (5): there is ", where". Although I am not a native English speaker, I must emphasize that English is excellent.
Taking all above into account, I strongly recommend acceptance of this draft into publication. I believe that the work done by the authors will be appreciated not only by the solar physics community but also by other astrophysicists and physicists.
Author Response
We are highly grateful to the reviewer for the recommended acceptance of the paper.